# Fuzzy Petri Nets for Traffic Node Reliability

**DOI:** 10.3390/s24196337

**Published:** 2024-09-30

**Authors:** Gabor Kiss, Peter Bakucz

**Affiliations:** Institute of Safety Science and Cybersecurity, Obuda University, 1034 Budapest, Hungary; bakucz.peter@bgk.uni-obuda.hu

**Keywords:** Petri nets, fuzzy analysis, minimum cut set, autonomous vehicles, real measured database

## Abstract

Self-driving cars are one of the main areas of research today, but it has to be acknowledged that the information from the sensors (the perceptron) is a huge amount of data, which is now unmanageable even when projected onto a single traffic junction. In the case of self-driving, the nodes have to be sequenced and organized according to the planned route. A self-driving car in Hungary would have to be able to interpret more than 70,000 traffic junctions to be able to drive all over the country. Besides the huge amount of data, another problem is the issue of validation and verification. For self-driving cars, this implies a level of complexity using traditional methods that calls into question the economics of the already existing system. Fuzzy Petri nets provide an alternative solution to both problems. They allow us to obtain a model that accurately describes the reliability of a node through its dynamics, which is essential in perception since the more reliable a node is, the smaller the deep learning mesh required. In this paper, we outline the analysis of a traffic node’s safety using Petri nets and fuzzy analysis to gain information on the reliability of the node, which is essential for the modeling of self-driving cars, due to the deep learning model of perception. The reliability of the dynamics of the node is determined by using the modified fuzzy Petri net procedure. The need for a fuzzy extension of the Petri net was developed by knowledge of real traffic databases.

## 1. Introduction

The idea of self-driving vehicles is not new. It emerged at the beginning of the 20th century, but it was only in the 1950s that serious steps in this direction began to be taken [1]. However, it is only today that major advances in hardware and computing power have made it possible to do so in a way that leaves room for passengers in the vehicle. The initial enthusiasm has set target implementation dates that are not sustainable. Fully self-driving vehicles (SAE Level 5) will have to wait, because, despite the progress in hardware, compliance with automotive safety standards means more control and thus more computationally demanding tasks. For this reason, the application of automotive safety standards to self-driving systems is currently difficult [2].

Driverless cars should have been on the roads in Germany for a long time. Engineers at almost all car manufacturers have been working on automated and highly automated driving systems for many years. However, the ambitious timetable for fully autonomous driving has been repeatedly postponed. The technology to be developed by the car manufacturers and the legal situation in terms of legislation were apparently more complex than expected.

In May 2021, the European Parliament approved a law allowing fully autonomous vehicles to drive on public roads in Europe. Work has been and continues to be carried out on the specific implementing provisions. Nevertheless, it is likely to be many years before the first driverless cars are on European roads. The transition phase will be characterized by highly automated driving situations: when parking cars in parking garages, in traffic jams, and when driving in convoys on the motorway.

For society, one opportunity for autonomous cars is to better integrate older people or people with limited mobility. They can get into the car and be taken wherever they need or want to go: whether to the doctor, to the shops, to visit friends, or to the opera. At the same time, each and every individual will be able to use their time in the car productively or for recreation thanks to the appropriate technology.

Traffic may run more smoothly and goods could be transported in a more rationalized and environmentally friendly way. Fully automated taxis or buses may one day be so cheap that rural areas can also be opened up.

One thing is certain in any case: the level of automation will further reduce the number of accidents. After all, human error is the cause of 90 percent of all crashes. However, this process will take a long time because conventional and automated vehicles will continue to drive in mixed traffic for many years to come. It is important to prevent the autonomous systems from failing or misjudging traffic situations.

The automotive industry has set a small milestone in this respect, with car-to-infrastructure communication (Car2X). With Car2X, vehicles can exchange helpful information about the flow of traffic or traffic obstructions and danger spots. The technology is already on board VW models at no extra charge. The Golf communicates via WLAN. Other manufacturers rely on mobile radio as a transmission channel.

The competition for the best technology for autonomous driving is in full swing worldwide. Development is the most advanced in the USA. Waymo—a sister company of Google—currently operates a fleet of 250 robotaxis in the city of San Francisco. Paying customers can use an app to order a car to their location, where they are then picked up by a Jaguar iPace that has been upgraded for autonomous driving and chauffeured to their destination completely independently.

The run on the Waymo offer is huge. So huge that it is illusory for new customers to be served. The vehicles’ capabilities are now so good that they even seem to be able to handle tricky traffic situations with oncoming traffic in roadworks or with cars stopping in the lane.

The technology for this is extremely complex and expensive. This is reflected in the fare, which is around the same level as a taxi journey with a human at the wheel. It remains to be seen whether the Google cars will prove themselves in the long term without revealing any safety gaps.

After all, there have always been curious cases that revealed technical deficits in previous pilot tests. Like in August 2023, when chaos broke out on the streets of the Californian metropolis. Several robotaxis from the company Cruise—Cruise is owned by automotive giant General Motors—blocked a road. The suspected cause was a data leak due to overloading of the mobile phone network. In another case, a robotaxi got stuck in the fresh concrete of a building site. Yet another time there was an accident involving a fire engine.

An accident at the beginning of October 2023 involving a driverless taxi had serious consequences. In the crash, a pedestrian was hit by another car and thrown in front of the robot taxi. The taxi’s emergency braking was unable to avoid a second collision. Tragically, the cruise car then drove to the side “for safety reasons” and dragged the woman several meters during the maneuver. The accident victim died [3].

In light of these circumstances, two solutions are possible for the development of self-driving cars. Either testing is performed on a test track for a given environment or traffic situation, or simulations are used to reduce testing costs. In simulation, for example, testing should be performed for all directions of a traffic intersection with very small changes in movement directions to match the dynamics of the intersection. It can now be seen that while simulation reduces the cost compared to testing under real conditions, there are still a large number of test situations and traffic conditions to be tested. Research results can accelerate the implementation of self-driving, which, while meeting safety requirements, can reduce the number of simulations to be performed, reduce the number of conditions to be taken into account, etc.

Amplifying safety, security, and reliability requirements necessitate the development of adequate methods and tools for modeling traffic flow in nodes.

Fault tree analysis (FTA) has been extensively used for these systems since the beginning of the 1960s and is approved by a large amount of scientific materials.

Fault tree analysis interprets issues as:-What are the motives for a disagreeable event?-How immense is the probability that this event will take place?

The fundamental concept in FTA is the translation of a physical system into a structured logic diagram, the fault tree, in which certain specified causes lead to a specified top event (TE) of interest.

With the basic fault tree analysis, basic and intermediate elements are transmitted through logical gates (generally “and” and “or” gates) using a tree topology resulting in a top event, which could be identified as the reliability of the system (see Figure 1).

However, the parameters affecting the traffic system are highly uncertain: the system components committed during the software processing chain have uncertain correlations. To deal with uncertainty, it would be appropriate to use fuzzyfied data and mathematical operation structures. Homologic (non-fuzzy) results cannot show the substantial setting of the real system operations.

A new approach using fuzzy FTA was recommended in order to address the Kuznetsov fault tree problems and the modern state of investigations [4].

The general technical aspects published in fuzzy FTA literature concentrate almost exclusively on nuclear engineering issues. Among others, the Nuclear Engineering Group of University Palermo published an open-source software, Treezzy2, where all procedures for top event reliability were coded in a software system [5].

Using the software, each basic event is considered a

Typ “double” value with FTA, and parallel toTrapezoidal fuzzy structure.

First the classical FTA cut set method is taken into consideration and then the same system with trapezoidal membership functions will be evaluated using the fuzzy FTA. Thus, it is conceivable that a system with many basic events can be investigated using the fault tree technique in real autonomous driving situations since there is a possibility of control.

Fuzzy FTA-based solutions have been introduced by the authors into research on self-driving cars. The idea is to replace the outage probability with fuzzy outages and to make the system reliability (the top event) a mixed fuzzy event [6].

In this paper, we would like to present a method to determine the real-time reliability of traffic node dynamics and to test an optimized (critical) minimum cut-set method for an embedded fuzzy FTA system.

Determining the quantified reliability based on the basic fuzzy events by the fault tree analysis is an important application area. However, the exact determination of the reliability of traffic nodes based on fuzzy events is not well established in the literature.

The advantages of the method: The Fuzzy FTA method, the top event, defines a metric to be able to compare diverse transport nodes to take into account the complexity of autonomous self-driving levels 4 and 5.The method simulates the real vehicle movement of the node with statistical distributions which are then integrated into fuzzy membership functions.By fuzzy operators, the traffic system uncertainty can also take into account.

However, there are several drawbacks to the fuzzy FTA process.

The cut-set solution to the fault tree calculation poses a number of convergence problems, which makes embedded real-time automotive applications not feasible.On the other hand, the applicability and recognizability of fuzzy basic events for engineers at the frequentist level pose problems. This is particularly true in self-driving practice, where the top event, i.e., the interpretation of system reliability, is now a mixed fuzzy function, which is difficult for many engineers to release, i.e., sell.When testing a traffic node, many redundant events are tested, which differ only slightly, e.g., a given vehicle turns a corner with a slightly larger curve than in the previous simulation. The problem with the implementation is that the number of test cases grows exponentially, which becomes unmanageable after a while. For example, a permutation of 6 fuzzy base events, by systematically changing the membership function leads to 720 possible fuzzy variants. For larger base event numbers, the problem is not tractable with current methods. As a complex solution to determine the reliability of fuzzy FTA, we develop a minimum cut set algorithm and use a small-scale model to test the results of the minimum cut set based on percolation and to verify the theoretical results, since the reliability of complex engineering systems typically depends on their dynamic behavior [7].

The Petry mesh system points the way to these major drawbacks.

Petri nets are much better suited than classic models of reliable systems for describing such processes [8,9,10]. The IEC 62551 standard (Analysis techniques for dependability-Petri net techniques) makes suggestions for this.

For sufficiently complex or not purely Markovian systems, only simulation can be used to calculate the reliability parameters. An open problem here, however, is the very long runtime for the statistically verified determination of very small (failure) probabilities. very small (failure) probabilities.

So far, very few projects have addressed the use of Petri nets in the analysis of the reliability of self-driving cars. The main reason for this is that the necessary complexity in the field of determining the reliability of the dynamics of a given traffic node could not be provided by the nets. This requires an extension of the traditional Petri net [11].

The proper functioning of dynamic diagnostic and recovery procedures, such as traffic node dynamics, requires relevant information about objects and resources, and their changing states. It is also necessary to be prepared for the possibility of unreliable data [8]. The fault tree, a traditional logic-based method, is not suitable for this purpose, as its structure makes it difficult to take into account any change. Petri nets are well suited for representing the state of a dynamic system of traffic nodes sharing resources and consisting of concurrently active objects, but they are not suitable for handling uncertainty. In order to handle uncertainty, we need to combine the theory of fuzzy sets and Petri nets to create a fuzzy Petri net, which can be used to represent uncertain knowledge about the reliability of the system state of traffic nodes [12].

The reliability of the transport node in autonomous driving means that the geometry and dynamics of the node are identified, analyzed and quantified using fuzzy Petri nets. 

In order to get the reliability of the traffic node, it is essential to determine the dynamic, load, and geometric complexity of the traffic and to create a metric that makes the nodes comparable to another traffic node dynamics.

In this paper, we present an algorithm developed with the aim of determining the reliability of a traffic node based on the estimation of fuzzy Petri nets.

A research group at the University of Óbuda is analyzing the safety and reliability of the system in a separate project.

The main milestones of the project are:Determining the reliability of the transport node based on the minimum cut set of fuzzy Petri netsCreating a fuzzy Petri net of a real traffic nodeCollecting traffic data of the traffic node, andAnalyzing the traffic node using fuzzy fault tree analysis of the Petri net.

In our paper, we focus on the first three milestones, as our task now is to determine the reliability of a traffic node using a fuzzy Petri net.

The reliability of a node means how we can use a quantity to describe the complexity of a node geometry and dynamics and what a “strong” complex algorithms (deep learnings) can be used to model and account for ominous node events.

In autonomous driving, determining the reliability of the transport node based on the modified Petri nets is an important application area. However, the exact determination of the reliability of traffic nodes based on fuzzy events together with Petri nets is not well established in the literature.

The advantages of the method: The modified fuzzy based Petri net method, defines a metric to be able to compare diverse uncertain transport nodes to take into account the complexity of driver assistance systems.The method simulates the real uncertain vehicle movement of the node with statistical distributions which integrated into fuzzy Petri net.By fuzzy operators, the traffic perception and planning system uncertainty can also take into account.

In particular, we emphasize that our publication is the first attempt in the practice of self-driving car design to establish the reliability of a complex traffic intersection system under embedded real-time conditions.

That is, in this paper, we do not present the results of a decade-long experimental system, but rather the application design of a brand new system.

After the basics of fuzzy algebra and Petri nets (Section 2), we present the reliability determination by minimum cut set method, using fuzzy Petri nets (Section 3). In Section 4, we present our first experiments in a self-driving ID Buzz as an egocar and look at the reliability of detecting the passage of a car using fuzzy FTA and fuzzy Petry Net. Finally, in Section 5 we present our first results and conclusions.

## 2. Fuzzy Analysis, Petri Nets

In our research, we aimed to model traffic flows at traffic intersections using appropriate tools and methods to mitigate the computational complexity explosion that currently used methods lead to, while maintaining safety and reliability requirements. Our starting point is the use of Petri nets, the analysis of which has been the subject of several research studies since the 1960s [13].

In order to be able to produce products with a certain level of reliability, the most accurate information about the failure behavior should be available. It is not possible to accurately predict the failure time of the components because they change over time. The lifetimes achieved can be analyzed using statistical methods and approximated using lifetime distributions. System simulation models can be used to examine the effect of component failure on system behavior. Application of fuzzy methods it enables reliability calculations that are even more in line with reality. The lifespan of a technical product depends on many properties. Tens and events that together affect reliability and others reflect the quality characteristics of the product. These properties and events They are diverse, partly interdependent and subject to change.

At the time of the introduction of Petri nets, the prediction of non-functional properties was necessary in many application areas before realisation. This is particularly difficult in the early design phases, as such properties usually only emerge from the interaction of all components (emergent properties).

On the other hand, they are just as important as functional requirements in safety-critical applications, especially if release-certification is mandatory. With the help of a model and suitable analysis methods, well-founded decisions can be made and the effect of design parameters can be analysed.

Add to this, that a number of successful models and methods are available for this purpose; in practice, static models such as fault trees are primarily used. However, the reliability of complex technical systems like autonomous driving can be improved by, among other things, fault tolerance measures such as reconfiguration or cold standby, whose effects can only be analyzed with their dynamic behavior.

In addition, the dynamic interaction of system behavior and control as well as a changing environment can have an influence on reliability of autonomous driving.

Without considering these properties, models and their statements become unrealistic or unnecessarily conservative, so that the system variant sought with the best cost-benefit ratio cannot be found.

Petri nets and their extension by stochastic time and fuzzy are very well suited to describing such complex autonomous driving processes. IEC 62551 (Analysis techniques for dependability-Petri net techniques).

Generaly an overview of various models and combinations of static and dynamic descriptions and their dynamic descriptions and their modelling power can be found in [14,15].

For simple dynamic models for autonomous driving with memoryless time behaviour, Markov chains can be used and numerically and analysed numerically [16]. For models with a large state space or models with non purely Markovian time distributions only simulation can be used to calculate the reliability parameters.

The latter is often the case in many technical systems due to clock cycles, deterministic maintenance intervals or Weibull–distributed lifetimes. An open problem here, however, is the very long runtime for the statistically validated determination of reliability parameters, as the events of interest only occur very rarely in the model. events only occur very rarely in the model. This problem is known as rare-event simulation (rare-event simulation), and in the literature, it is mainly analysed using techniques of importance sampling or splitting [17].

What these methods have in common is that the events of interest events are forced in the simulation in order to gain more significant information from the same number of simulated events. The results must be suitably converted in order to minimize the changes that distort the normal simulation process. Splitting methods and the fuzzy algorithm in particular, have the advantage of being less dependent on the simulated model.

They are easier to automate and therefore more suitable for implementation in a software tool software tool that can also be used by system developers without detailed background knowledge of the mathematical relationships.

Current research work in the Safety Engineering working group at Obuda Univesity is focusing on the simulation of complex [18], reliable systems [11], which can be accelerated by several orders of magnitude by exploiting the structural properties of Petri nets and can be accelerated by several orders of magnitude [19,20].

In our fuzzy based Petri system is briefly shown how the models proposed in IEC 62551 models of typical reliability patterns proposed in IEC 62551 can be simplified with standard model extensions.

The formal way to describe the flow of processes in systems are the so-called Petri nets (named after their inventor Carl Adam Petri). Note that the term “system” includes not only computers, but any organizational, technical, and computer-based system in which controlled flows of objects and information are important. A Petri net is a labeled graph in which nodes symbolize processes and their associated states and edges symbolize flow conditions. In Petri nets, a distinction is made between 2 types of elements:-static elements, which serve to represent the structure, and-dynamic elements, which serve to simulate the processes within the model.

Petri nets consist of 3 types of static elements: On the one hand, there are the places, which are also called S-elements, states, conditions or places. They describe the “current situation of the system” and are represented by circles. On the other hand, there are transitions, which are also called T-elements, events, or actions. These are symbolized by rectangles and represent the transition conditions between the locations. Finally, there are the directed edges, represented by arrows. These edges connect us to transitions and show the course of the process through the direction of the arrow. A position is always followed by one (or more) transition(s), followed by another (or more) position(s), but a position never follows a position, or a transition follows another transition. The dynamic elements in Petri nets are called brands (or markings). They can be located in positions and transitions and indicate the state of the system. In their simplest form, marks are indistinguishable from each other and are taken from their input point(s) and generated in their output point(s) when a transition is switched (Markers are not moved in a Petri net but are removed and created!). There are 3 different types of Petri nets: B/E systems (condition/event), S/T systems (position/transition) andPr/T systems (predicate/transition).

B/E systems are the simplest form of Petri nets. Each location can only have a maximum of one brand, whereby all brands are the same (i.e., anonymous) and therefore cannot be distinguished from one another. The marking of the sites can be done exogenously or endogenously in B/E systems. With exogenous markings, the marked situation is specified from outside, i.e., by the model user, who sets the marks, for example, based on empirical observation or based on a test specification. Endogenous markings are created by “switching on” the initial markings.

A distinction is made between input and output states: Input states (or preconditions) are the conditions (positions) immediately before an event (transition).Initial states (or postconditions) are the conditions immediately following an event. 

S/T systems (positions/transition networks) are used to represent complex processes because they allow more than one mark to be placed in a position and thus allow the processes to be viewed quantitatively. Although the brands are independent, they are still anonymous (i.e., cannot be distinguished from one another). A capacity is set for each location, which indicates the maximum number of brands it can hold. If there is no capacity information at one point, then by definition, it is assumed that the capacity is infinite. Edge weights are defined for the edges, which indicate how many marks are removed or inserted when the transition is switched. If no edge weight is specified, the weight of the edge is set to 1. In order for the transition to switch, there must be at least as many markers in the preconditions as would be removed during switching. There must be enough free capacity in the postconditions to be able to generate all brands. However, it is no longer necessary to remove all marks from the prerequisites. Postconditions do not necessarily have to be empty either.

In Pr/T systems, the brands are individual or colored, i.e., they are distinguishable and may have certain properties. You can now also specify switching conditions at the transitions so that switching can only take place when certain brands are present. This is necessary in order to be able to model more complex processes completely and in detail.

During switching processes in Pr/T systems, you can see which subsequent state was generated by switching with a specific brand. In the context of Petri nets, a conflict refers to ambiguous behavior of a system. This means that 2 events compete for a condition, both are activated, but only one event can take place because the occurrence of one no longer activates the other.

Conflicts can: -Oexogenously, by specifying a decision rule, or-Ondogenously, by setting up a so-called regulatory circuit.

The following 3 types of conflict can be distinguished in B/E systems: 

Branch conflicts. Two (or more) transitions branch from a marked point, but not all of them can be switched (since after switching one transition the other would no longer be activated). This problem can be solved by an internal control loop.

Competitive conflicts. These arise when two (or more) transitions want to switch to the same postcondition. This problem can also be solved with an internal control loop.

Confusion. This is a combination of the two types of conflict: branching and competition conflict. In order to solve this problem, the model must be expanded to include a control loop or externally specified decision rules.

Petri nets alone can only be used to a limited extent to model the reliability of transport nodes because the components of the node dynamics are largely uncertain, and therefore the use of rigid graph elements is not practical.

Since uncertainty plays a significant role in determining reliability in the present case, and the transport node is not the classic case of determining the reliability, in our article we use a combination of Petri nets and fuzzy arithmetic.

To cover the traffic dynamics, it is necessary to discretize the node, as well as to cover and model the dynamics of the Petri nets.

In order to account the uncertainty, it is necessary to record the fuzzy arithmetic in the Petri net, which is realized by taking into account the discrete traffic dynamics (tracking certain vehicles and setting up statistics) Petri net-tokens as fuzzy membership functions.

For the reliability of node dynamics, we need to construct the Petri net of the node, which is represented in the article as follows.

## 3. Determining the Reliability

The evaluation and validation of passive safety systems is based on a practicable number of crash tests under defined test conditions. The procedure is established and recognised worldwide. When testing systems for assisted and automated driving, problems arise on the one hand from the large number of relevant scenarios and from the variability in the interaction between driver, vehicle and environment. On the other hand, methodological limits are encountered especially when testing sensors and algorithms in the field (artefacts, misinterpretation of traffic situations, “false positive” reactions of the systems).

The current (and planned) EuroNCAP tests of driver assistance systems are suitable for making systems comparable from a consumer protection perspective. They do not take sufficient account of the aforementioned problems and methodological limitations. In addition, they may force manufacturers and system developers to optimise the systems to meet the test criteria and not to achieve the greatest possible effectiveness for road safety. They are therefore hardly suitable for evaluating the effectiveness and validating the systems [20].

Reliability of the traffic node means that the geometry and dynamics of the node are analyzed using a modified Petri net method. The membership function of fuzzy token distribution is determined by defining the dynamics of the node.

The authors therefore propose an integrated concept for evaluating the effectiveness and validation of assisted and automated driving systems. It consists of various coordinated test tools. The relationship between the scenarios and the test tools is illustrated in the following figure with the aid of the V-model [21].

Laboratory tests and simulation are specified on the basis of the scenarios to be considered. The strength of laboratory tests and simulation lies in the fact that they make it possible to map the variety of scenarios and the variability in the interaction between driver, vehicle and environment. On the other hand, assumptions have to be made and Laboratory tests and simulation are specified based on the scenarios to be considered. The strength of laboratory tests and simulation lies in the fact that they make it possible to map the variety of scenarios and the variability in the interaction between driver, vehicle, and environment. On the other hand, assumptions have to be made and more or less simplified models have to be used.

The results of laboratory tests and simulation are used to specify the proving ground tests. Conversely, the assumptions and calculation models of the laboratory tests and simulationS are verified with the help of the proving ground tests. The strength of the proving ground tests lies in their reproducibility through automation. This makes it possible to statistically validate the results of laboratory tests and simulation. In addition, critical manoeuvers that are not possible or desirable in field tests can also be performed in a controlled environment. On the other hand, the variety of scenarios and the variability in the interaction between driver, vehicle and environment cannot be modelled on the test site and the results cannot be easily transferred to the field [22].

In the end, field tests therefore ensure that the assumptions and hypotheses made are accurate and complete. By running defined scenarios, it is validated that the interaction between driver, vehicle and environment in the field is as assumed. It is not necessary to drive arbitrarily long distances or long journey times [23,24,25,26,27,28].

Each individual tool has specific strengths and weaknesses. By working together within the toolchain, the strengths are utilised and the weaknesses has been compensated for. The tool chain is therefore suitable for effectively evaluating and safeguarding systems for assisted and automated driving. more or less simplified models have to be used.

In the present case, fuzzy Petri nets are used because the number of testing possibilities is limited, and it may also be necessary to take rare events into account.

The first step in the application of these nets can be to discretize the space and record its topology by recording the vehicle movements of a given node at intervals. The vehicle motion determines primarily the fuzzy nature and applicability of Petri nets.

By recording a traffic node, e.g., with a drone or similar, we can record the dynamics of the node (Figure 1).

It is very important to form a graph where the movement of vehicles between nodes is interpreted.

Once the dynamics of the node is captured, the next step is to break the ice for the formalism required for fuzzy Petri nets, ultimately to determine the reliability of the transport node in real-time embedded systems → The reliability could be defined as the mean, steady state variance of the deviation from all node values of the Petri nets.

For the fuzzy Petri environment, now we have to define a reward Function (1):(1)$$r_{m}=\frac{1}{N}\sum_{\iota =1}^{N}\underset{t\to \infty }{\lim}{weight}_{i}\left(t\right)-\frac{1}{N}\sum_{j=1}^{N}{\mathrm{weight}}_{j}\left(t\right)$$

The values of the reward are (2):(2)$$r_{m}=\left\{\begin{array}{c} 1\;if\;M\;\in \;O\\ 0\;if\;M\;\in \;F \end{array}\right.$$
where *r_m_* is a state reward, which splits the set of reachable markings of a fuzzy traffic node Petri net into two subsets: O represents the operational state of the system, andF represents the failure state.

A traffic node failure status means that the ego car wants to move to a position that is not allowed. For example, if the transition to the right-hand neighbor of the discrete cell element is allowed, then 1 if it is not allowed because e.g., sidewalk is there, then 0.

Remember the node-edge formalism: emphasize that the traffic node is represented by a graph, where the dynamics take place along the edges of the graph, i.e., between two selected geometric points, each vehicle or pedestrian passes through per given unit of time (Figure 2). The elements of the Petri net (P V and tokens and firings) represent the traffic rules, traffic signs, and detection knowledge interpreted at the node. The place nodes signed as p and the transition nodes are signed as t. E W S, and N are the directions (like east, west, south and north). The token distributions are fuzzy membership functions. The reliability of the traffic node based on fuzzy Petri net.

The instantaneous steady state and the interval availability metrics can be calculated using this state reward function.

However, a key element of the method is to approximate the movement of vehicles with a fuzzy system and to put this fuzzy property into the formalism of a Petri net to determine the reliability of the node.

By discretizing the geometry of the node, we define neighborhood relations between each element using a cellular automata formalism, i.e., a center element as shown in Figure 1 has 8 neighbors (1, 2, 3, 4, 5, 6, 7, 8) and it is from these positions that the vehicle travels through, for a given number of positions and time units.

The statistics can then be plotted on a graph (see Figure 3) to determine the relative frequencies. The relative frequencies are the number of cell elements summed with the center on the *x*-axis and the number of vehicles that passed from the directions shown on the *x*-axis to the center in a unit of time on the *y*-axis.

The most important part of the paper is that these relative frequencies are then considered as the fuzzy number, which is incorporated into the Petri net formalism, much like the distribution of the token count of a net (see Figure 3, Steps D and E).

How can we summarize our method? First, we found that in the case of self-driving cars, the information from the sensors (the perceptron) is a very large amount of data that is now unmanageable for a traffic junction. In self-driving, nodes need to be ordered and systematized. In Hungary, for example, a self-driving car would have to interpret more than 70,000 traffic junctions.

The other problem is the issue of validation and verification. For self-driving cars, this implies a level of complexity, if conventional methodologies are used, that fundamentally questions the economics of the system.

Therefore, an alternative solution was found in fuzzy Petri nets.

The fuzzy nature is represented by the dynamics of the traffic node, by producing local statistics based on drone images and, for a 3 × 3 matrix, by considering the frequency diagram of the movement of vehicles from cell to cell as a fuzzy number.

The Petri net nature was prescribed by the real-time embedded system to compute the reliability of the whole system: we put the fuzzy distributions into the token count distribution of the Petri net.

This gives a model that exactly describes the reliability of a given transport node by its dynamics, which is essential in perception.

The more reliable a node is, the smaller the deep learning mesh required.

Based on our intensive measurements, we concluded that if *r_m_* ≥ 185 the traffic node is reliable.

The following steps are necessary to realize the determination of the modified Petri net-based reliability of a traffic node.

A → Let’s take a node and a traffic situation to be modelled,

B → Discretizing the node with a rectangular grid (cells), 

C → The neighborhood relations be determined for each cell and for each time stamp. 

D → Relative frequencies (cell probability density) after n time stamps (numerical experiments) here for example Crossing cell numbered with 1 → to cell numbered with X: 0 times, 

crossing cell 2 → X: 3 times,

crossing cell 3 → X: 2 times,

crossing cell 4 → X: 4 times

crossings cells 5 6 7 and 8 → X 0 times

E → From the relative frequencies a fuzzy membership function is created for basic event 1 is created. 

Go To A → and a new traffic situation is taken into consideration. If we have more than 300 examples (traffic situations) and the integrated relative frequencies are then calculated:

F → The final fuzzy membership function as a token count distribution for Petri nets.

G → Traffic flow graph to be created based on real experiments.

H → The traffic flow graph is now a Petri net with token distributions interpreted on the nodes. Analysis of the Petri net is the reliability of the traffic node based on real traffic experiments.

## 4. Experiments with Autonomous VW ID.Buzz

The first real-world experiments were carried out on an airport runway, detecting the passing of a car and using an ID.Buzz as an egocar (Figure 4). The system shows the passing of the car with an approximate step function increase in reliability.

This was carried out using the fuzzy FTA system (Figure 5) and the Petri net system.

In the experiment, channels 5 and 7 of the VW ID Buzz radar sensor were used as the base events in the fuzzy FTA and as the input data of the Petri net (Figure 6).

The autonomous car processes five gigabytes of data per minute as a basis for making driving decisions. The computing power on board is roughly equivalent to that of 15 laptops. Future vehicles should be able to predict traffic events for around 10 s in advance and master all possible traffic scenarios, anywhere in the world. And the systems of the future must not only be developed to be traffic-proof, but also data-proof in order to fend off potential cyber-attacks. The figures show the major challenges facing car manufacturers.

Volkswagen’s development partnership with the American tech company Argo AI was quickly terminated, but the idea of autonomous vehicles was not shelved. The prototypes were based on the all-electric ID.Buzz is currently being further developed with Mobileye. Equipped with a combination of lidars, radars, cameras, and laser scanners, test drives with prototypes are taking place in Munich.

In the experiment, the egocar is to detect the reliability of the ID.Buzz and the passing of a vehicle (Figure 7).

In the figure, we have plotted the vehicle passing process and at the bottom of the figure the reliability of vehicle detection using fuzzy FTA and fuzzy Petri Net.

It can be seen that the two detection reliabilities are identical.

This was carried out for several other experiments and similar results were obtained. 

The major difference is that the Landa notation of fuzzy FTA is o(n^5^), while that of fuzzy Petri Net is o(n^2^), i.e., it is more expedient to use in real-time embedded systems.

## 5. Discussion

In this work, we presented a sub-project of the autonomous management project of the University of Óbuda and engaged with the fuzzy Petri Net-based determination of the traffic node reliability for autonomous driving perception deep learning issues.

Determining the reliability of transport nodes based on the modified Petri nets is an important application area, however, the exact determination of the reliability of traffic nodes based on fuzzy events together with Petri nets is not well established in the literature.

The advantages of the method: The modified fuzzy-based Petri net method, defines a metric to be able to compare diverse uncertain transport nodes to take into account the complexity of driver assistance systems.The method simulates the real uncertain vehicle movement of the node with statistical distributions which integrates fuzzy Petri net.By using fuzzy operators, the traffic perception and planning system uncertainty can also be taken into account.

In our paper, we also presented the flow chart of our new system with eight points for a real-live node in Szolnok, Hungary, and the result was estimated to be 1.14 × 10^−8^ ppa for the reliability.

Next steps in modelling:Recording the traffic dynamics of the selected traffic nodeConstructing the fuzzy Petri nets and using them to calculate the reliability.Creating a small-scaled model and sensor set and performing traffic experiments.Comparing and scaling the results.

## Figures and Tables

**Figure 1 sensors-24-06337-f001:**
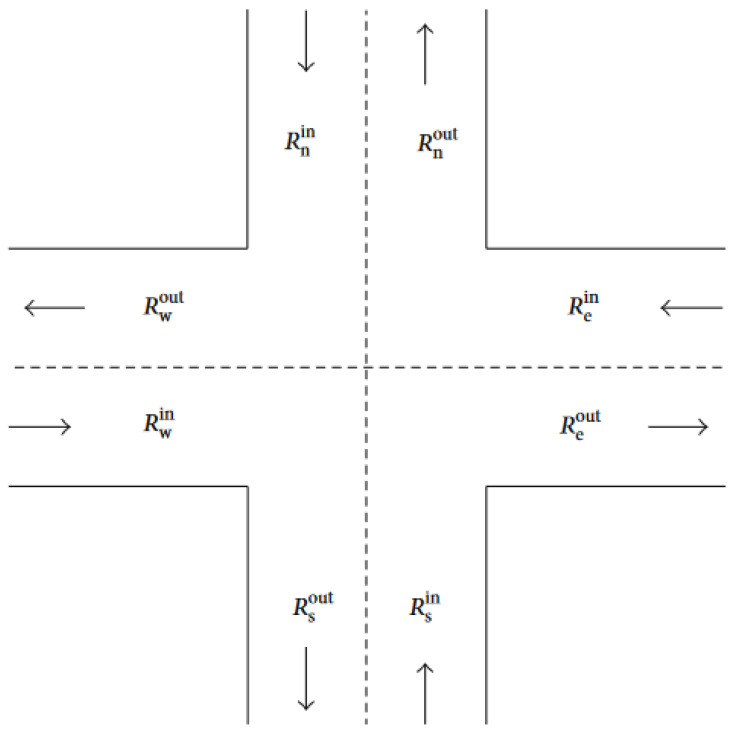
Traffic node to be used for the modified Petri net simulation (R means the traffic flow input and output (in, out) according to four equator (s, w, e, n)).

**Figure 2 sensors-24-06337-f002:**
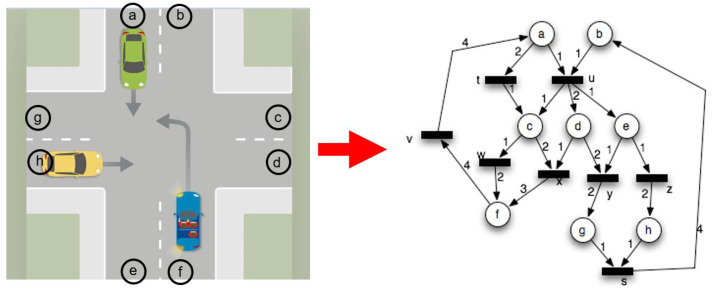
Petri net representation of our traffic node in Figure 1.

**Figure 3 sensors-24-06337-f003:**
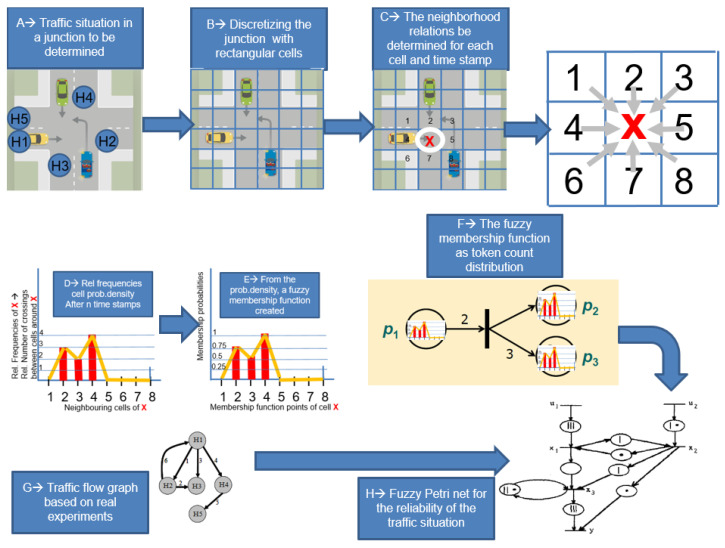
Flow chart of the system.

**Figure 4 sensors-24-06337-f004:**
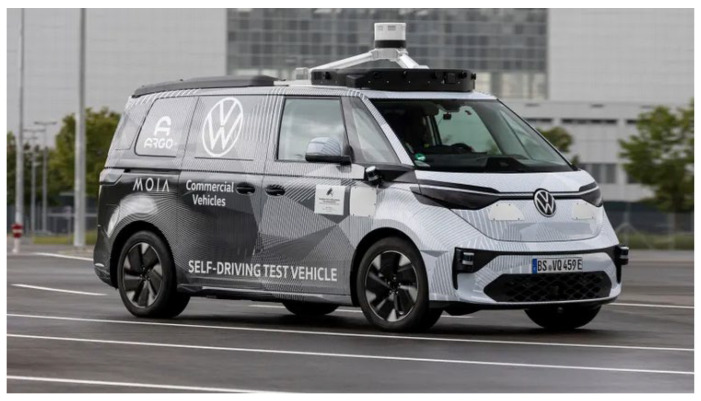
Testing the traffic node reliability with VW ID.Buzz.

**Figure 5 sensors-24-06337-f005:**
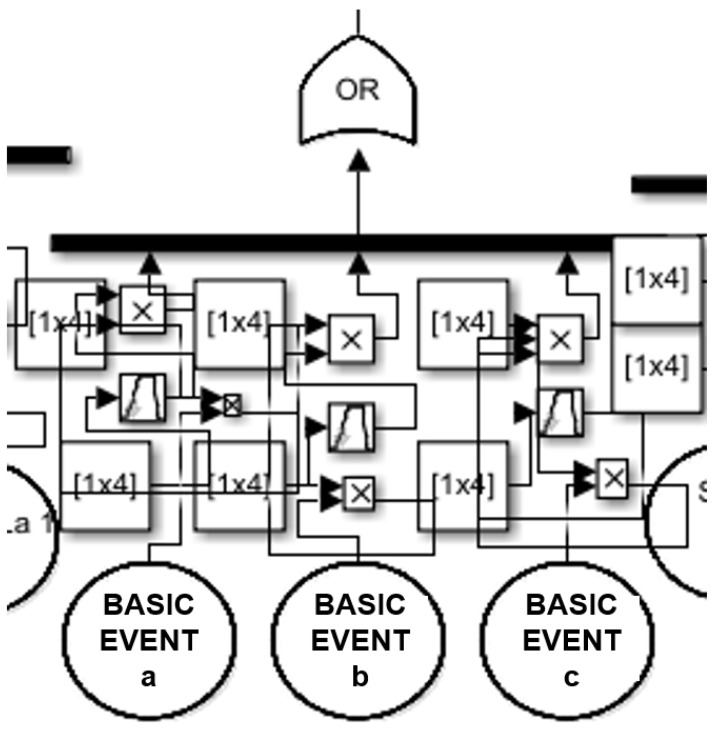
Usd fuzzy FTA for testing and comparing the Petri net results for the traffic node reliability with VW ID.Buzz.

**Figure 6 sensors-24-06337-f006:**
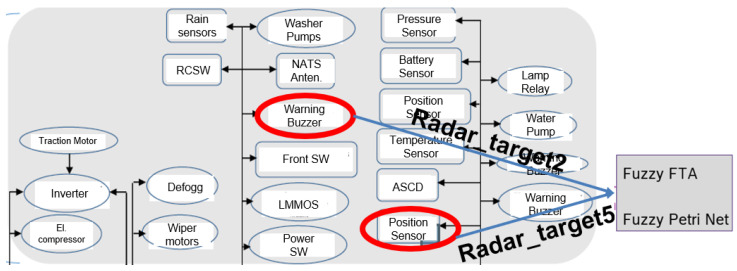
Perception system of the ID.Buzz testing the fuzzy FTA and Fuzzy Petri Net system Testing the traffic node reliability with VW ID.Buzz.

**Figure 7 sensors-24-06337-f007:**
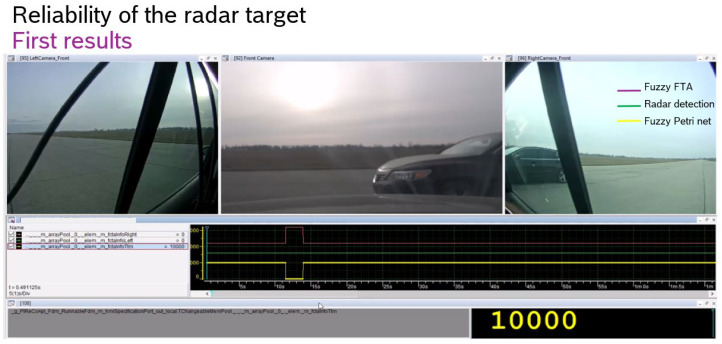
In ID Buzz as an egocar and a car passing through, we look at the reliability of its detection using fuzzy FTA and fuzzy Petry Net.

## Data Availability

Data are contained within the article.

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
