# Peer review of "Fuzzy Petri Nets for Traffic Node Reliability"

_sensors, 2024, doi:10.3390/s24196337_

Round 1

Reviewer 1 Report

Comments and Suggestions for Authors

1-      The paper should include more simulation results and figures.

2-      The work should include the nodes written in the discussion section:

         -Recording the traffic dynamics of the selected traffic node

         -  constructing the fuzzy Petri nets and using it to calculate the reliability.

         -  creating a small scaled model and sensor set and performing traffic experiments.

         - Comparing and scaling the results.

3-      Authors did not refer how their work differ from others.

4-      Finally, please make sure your discussion section underscore the scientific value added of your paper, and/or the applicability of your results, as indicated previously.

Comments on the Quality of English Language

Language not bad, but may be improved

Author Response

  1. No, because it is a brand new (patented by us in 2o23 (Bakucz, P., and Hruschka, M. Method for checking a reliability of a model of traffic dynamics at a traffic junction, 2021, patent: DE102021207629A1) 
    a system that solves the convergence problems of fuzzy fta and thus 
    and with it the limitations of real-time embedded system sub-analysis. That is, our paper does not a decade of research, but rather a quasi system design, how the world of fuzzy fta can be used to the case of traffic flows in the area of reliability for self-driving systems.
  2. Supplemented with the results of testing with ID.Buzz. 
  3. The paper have nothing to differ from others, because this is the first attempt in this particular field.
  4. The scientific strength of the paper is very strong, as it is the first attempt to continuously communicate the reliability of a self-driving system in a real-time embedded system, without having to generate thousands of fuzzy error trees in e.g. 0.2 seconds in a during emergency braking. 

Reviewer 2 Report

Comments and Suggestions for Authors

1. What is the main question addressed by the research?

The article examined aims to analyze the safety of a traffic node using Petri nets and fuzzy analysis to obtain information on the node's reliability, which is essential for modeling autonomous vehicles, thanks to the deep learning perception model.

2. What parts do you consider original or relevant for the field? What specific gap in the field does the paper address?

The innovative element lies in the use of Fuzzy Petri nets to solve the proposed task.

3. What does it add to the subject area compared with other published material?

The article does not clearly demonstrate what the introduced innovation is, as there is no chapter on the state of the art. It is suggested to integrate such a section to allow for a comparison.

4. What specific improvements should the authors consider regarding the methodology? What further controls should be considered?

It is suggested to also consider deep learning approaches such as Graph Neural Networks (GNNs), which are very relevant for the problem to be solved.

5. Please describe how the conclusions are or are not consistent with the evidence and arguments presented. Please also indicate if all main questions posed were addressed and by which specific experiments.

The work has a significant gap, that of the lack of experimentation. This is a major problem as the conclusions are not robustly supported.

6. Are the references appropriate?

The references are correct but are very few; it would be necessary to extend the state of the art.

7. Please include any additional comments on the tables and figures and quality of the data.

The figures are very large and of low quality; it would be necessary to resize them and redo them. Finally, equation 2 should be written out and not inserted as a figure.

Comments on the Quality of English Language

The text should be reviewed; the quality of the English is not very high.

Author Response

3. This is the first attempt to present a stable convergent system for Petri nets in a system of convergence node reliability, i.e. deep learning learnability. 

4. The task is not the development of deep learning or GNN, that is the job of the perception department of the car companies. Our task in this in this paper was simply to help improve reliability by taking into account the dynamics of traffic intersections real-time embedded system capture of reliability. 

5. a brand new (patented by us in 2o23 (Bakucz, P., and Hruschka, M. Method for checking a reliability of a model of traffic dynamics at a traffic junction, 2021, patent: DE102021207629A1) a system that solves the convergence problems of fuzzy fta and thus and with it the limitations of real-time embedded system sub-analysis. That is, our paper does nota decade of research, but rather a quasi system design, how the world of fuzzy fta can be used to the case of traffic flows in the area of reliability for self-driving systems.

6. the references are appropriate

7. we did.

Round 2

Reviewer 1 Report

Comments and Suggestions for Authors

The new revised paper is good

Comments on the Quality of English Language

The language is good and acceptable, but it can be improved

Author Response

Dear reviewer,
thank you very much for your work and your positive review.
We have tried to correct the English errors. We have tried to describe the methods as well as possible.
We trust that you will find our scientific work acceptable.

Reviewer 2 Report

Comments and Suggestions for Authors

Thank the authors for answering the questions. The authors clarified the unclear points.

However, the experimental phase is still too confusing. Is it possible to provide a qualitative/quantitative metric to verify the effectiveness of the proposed method?

Finally, the images need to be redone as they are low-quality.

Author Response

Dear reviewer,
thank you very much for your work and your positive review.
We have tried to replace the more difficult to see images with more appropriate ones. 
We have tried to describe the methods as precisely as possible.
We trust that you will find our scientific work acceptable.